# Factors in Infancy That May Predict Autism Spectrum Disorder

**DOI:** 10.3390/brainsci13101374

**Published:** 2023-09-27

**Authors:** Mina Gurevitz, Gerry Leisman

**Affiliations:** 1Well Baby Clinic Physician, Maccabi Health Services, Herzliya 4649713, Israel; mina.gurevich@gmail.com; 2Movement and Fetal Cognition Laboratory, Department of Physical Therapy, University of Haifa, Haifa 3498838, Israel; 3Department of Neurology, University of the Medical Sciences of Havana, Havana 11600, Cuba

**Keywords:** autism spectrum disorder, early intervention, obesity, autism biomarkers, head circumference

## Abstract

The global increase in the prevalence of ASD (Autism Spectrum Disorder) is of great medical importance, but the reasons for this increase are still unknown. This study sought to identify possible early contributing factors in children who were later diagnosed with ASD. In this retrospective cohort study, postnatal records of 1105 children diagnosed with ASD were analyzed to determine if any signs of ASD could be found in a large database of births and well-baby care programs. We compared the recordings of typically developing children and analyzed the differences statistically. Rapid increases in weight, height, and head circumference during early infancy predict the development of ASD. In addition, low birth weight, older maternal age, and increased weight and height percentiles at six months of age together predict the development of ASD. At two years of age, these four parameters, in addition to impaired motor development, can also predict the development of ASD. These results suggest that the recent increase in ASD prevalence is associated with the “obesity epidemic” and with recommendations of supine sleeping to prevent Sudden Infant Death Syndrome, associated with atypical neural network development in the brain.

## 1. Introduction

Autism spectrum disorder (ASD) is a complex neurodevelopmental condition characterized by cognitive, behavioral, and social dysfunctions. Autistic children experience severe difficulties in social interactions and communication, accompanied by limited interests and stereotyped repetitive behavior. ASD onset occurs at early childhood and often results in severe lifelong impairments [1], and thus is currently regarded as one of the most common childhood morbidities. Although the overall ASD prevalence is 23 per 1000 children aged eight years (one in 44) [2] and constantly rising, the etiology of its development and underlying triggers are still unclear. ASD is considered a multifactorial disorder elicited by genetic [3,4] and environmental prenatal, perinatal, and postnatal factors [5,6,7]. This large spectrum of putative triggers could derive from variations in methodologies (e.g., case definition, comparison groups, race and region, sample size, and exposure assessment) [8]. Adding to the complexity is the fact that ASD is a heterogeneous neurodevelopmental disorder with a broad range of clinical presentations (e.g., delays in motor development, in speech and language; lack of eye contact; limited socializing ability) [9].

Despite the need to identify a developing disorder during infancy, when brain plasticity enables manipulations and possible repair, ASD is mostly diagnosed in older children or in teenagers [10]. Evidently, early detection of signs that may predict later development of ASD remain a challenge due to clinical and practical implications [11], which improve prognoses [12]. While parents of ASD children have often noticed developmental problems prior to their first birthday, the only approved screening tool that identifies infants at risk is M-CHAT-R/F, available under the age of two years [13]. Barbaro and colleagues [14] have shown that the Social Attention and Communication Surveillance-Revised (SACS-R) and SACS-Preschool (SACS-PR) tools could be employed with high diagnostic accuracy for the identification of autism in a community-based sample of infants, toddlers, and preschoolers, indicating the utility of early autism developmental surveillance from infancy to the preschool period. Numerous reviews of early detection screening tools have, however, indicated equivocal degrees of effectiveness as primary early developmental variables have not been clearly delineated [15,16,17].

Although the core deficits of ASD involve social functioning [18], retrospective studies indicate that ASD children exhibit disruption in other developmental domains during their first-year postpartum, including motor, attention, and temperament [19]. Since retrospective studies are based on information derived from medical records, parent recall, and systematic observational recordings of home videotapes taken during the first or second year postpartum, and prior to a diagnosis of ASD [20], prospective longitudinal studies might provide a better description of the early symptoms of a forthcoming ASD and timing of the appearance of frank symptoms. However, most prospective studies thus far have focused on infants at high risk or who have an older sibling diagnosed with ASD. These studies indicate that the development of motor, cognitive, language, and social domains were grossly intact at six months of age, yet, a slowing in development was observable when nearing their first birthday [21,22].

In a study on infants at risk for ASD, abnormal postural control was already documented at six months of age [23], yet no single factor that could predict the development of ASD had been identified. These results prompted Wang et al. [24] to examine the magnitude of prenatal, perinatal, and postnatal variables on ASD development in a meta-analysis. They collected data from 37,634 autistic children and 12,081,416 typically developing children (TDC) (control children) enrolled in 17 studies and suggested that during the prenatal period, the risk factors associated with ASD were maternal and paternal age (≥35 years), race, gestational hypertension, gestational diabetes, maternal and paternal education, threatened abortion, and antepartum hemorrhage. During the perinatal period, the factors that were putatively associated with ASD development were caesarian delivery, gestational age ≤ 36 weeks, parity ≥ 4, breech presentation, preeclampsia, and fetal distress. During the postnatal period, the risk factors were low birth weight, postpartum hemorrhage, male gender, and brain anomaly. Then, they examined all of these factors individually but were unable to conclude whether they were causal or secondary in the development of ASD.

In a study seeking risk factors in early infancy that could predict forthcoming neurobehavioral disorders (NBDs) [25], analysis of the electronic health records (EHR) of 161 infants, who were later diagnosed with ASD, developmental coordination disorder (DCD) and attention deficit hyperactive disorder (ADHD), indicated deviations from trajectories on seven parameters (gestational age, birth weight, head circumference percentile, weight percentile, gross motor development, and difficulties in speech and communication). Collectively, these seven parameters have been suggested to predict forthcoming ASD with 85% probability. Most recently, Engelhard and colleagues [26] demonstrated that EHR during the first 30 days of life identified 45.5% of the ASD children, while similar analysis during the first year of life identified 60% of the ASD children. On the basis of these results, the authors have suggested that such an automated approach integrated with caregiver surveys could improve the accuracy of early autism screening.

The well-organized infrastructure of the well-baby clinics in Israel was exploited to examine the files of 1105 ASD children. The aim of the study was to derive detailed information from the Big Data repository of Maccabi Health Organization (MHO) in Israel to statistically analyze and compare the data obtained with that from TDC from the same population.

## 2. Methods

### 2.1. General Information

All neonates, infants, and children in Israel are routinely followed in well-baby care clinics (‘Tipat Chalav’) until five years of age under the supervision of the Ministry of Health, and thus their health profiles are documented. Maccabi Health Services (MHS), a state-mandated health maintenance organization in Israel (2.5 million enrollees), services 26% of Israel’s population. While recommended, the is no legal requirement for children to attend although most do on a regular basis, with parents making appointments but reminders being sent as well. The electronic medical records integrate data from the MHS central laboratory, medication prescriptions, records of purchases within the MHS pharmacy network, consultations, hospitalizations, procedures, and sociodemographic information. The socioeconomic status is based on a score of 1 (lowest) to 10, built for commercial purposes using geographic information systems and data such as expenditures of retail chains, credit cards, and housing. This score is correlated with socioeconomic status values provided by the Central Statistics Bureau [27] and is categorized into low (1–4), medium (5–6), and high (7–10) status.

### 2.2. Statement of Ethics Approval

The study was performed using patient charts and was approved by the Ethics Committee of Maccabi Health Services, Israel. No consent form was required by this Committee. The approval number is 0011-20-MHS.

### 2.3. Participants

A total of 2210 infants born between 1 January 2000 and 31 December 2017 were tracked from the EHR of the Pediatric Big Data resource of the MHS in Israel. The study group comprised 1105 children diagnosed with Autism Spectrum Disorder (ASD) between birth to 5 years (1 January 2000 to 30 June 2020). All children were diagnosed in accordance with the DSM-IV criteria (American Psychiatric Association, DSM-5 Task Force, 2013) [1]. Inclusion criteria in the study were as follows: gestational age (GA) 37–41 weeks (no premature) and at least 5 measurements of weight percentile (Wp) and head circumference percentile (HCp) available in the database; the control group of TDC comprised 1105 children from the same database, whose year of birth, gender, and local region matched those of the ASD children. The pediatric charts of the controls were carefully evaluated to ensure lack of ASD and then reviewed at time points identical to those of the study group. Most children (81%) were from families of medium–high socioeconomic level in the Sharon, South, Jerusalem, and Shfela districts in Israel. These data are summarized in Table 1.

### 2.4. Covariates

The following pertinent data were collected: (A) distribution to Maccabi districts and social status of family; (B) maternal age and alcohol consumption, cigarette smoking, and medications used during pregnancy; (C) child’s gender, gestational age (GA) and birth weight (BW); (D) biometric parameters: weight, height and head circumference percentiles (Wp, Hp and HCp, respectively) of child; (E) child’s developmental milestone achievement rates, including motor, speech, language, and communication (a developmental delay was assumed based on a letter of referral to a specialist, such as physiotherapist, speech therapist, occupational therapist, child neurologist and/or child psychiatrist) and the age of the child at the referral date was documented; (F) the first diagnosis made at the child developmental center before the final diagnosis of ASD and the age at first diagnosis; (G) age of the child at the time of ASD diagnosis; (H) additional medical diagnoses.

### 2.5. Statistical Analysis

Qualitative variables are described using incidence and percentages. Sequential variables are described by means and standard deviations. Normally distributed quantitative variables were analyzed using the Student’s *t*-test, categorical variables were analyzed at baseline using Pearson’s chi-squared test. A stepwise logistic regression model was used to examine the predictive added power of the independent parameters at each age to assess forthcoming ASD, and significant covariates from baseline analyses were added to the model. The statistical analyses were performed using SPSS (Version 28) software. The statistical significance was defined at *p* < 0.05.

## 3. Results

### 3.1. Demographic Parameters

Prenatal and perinatal demographic parameters of the ASD children were compared to those of the controls (TDC; Table 1). The ratio of male to female was 4.8:1 (higher than the ratio published in a systematic review of 54 studies [28], which has suggested that ASD is probably underestimated in females. From the data presented in Table 1, the only parameters that differed significantly between the ASD and TDC groups were maternal age and birth weight (BW). The average BW of the ASD children was significantly lower than that of the TDC (3.2764 ± 0.46 Kg vs. 3.32 ± 0.43 Kg; *p* = 0.007). Unexpectedly, no significant differences between the ASD and TDC groups were observed in parameters such as gestational age, gestational diabetes, exposure to cigarette smoking, alcohol abuse, and fertility treatments. This could have resulted from a lack of complete information and detailed documentation of these parameters in the EHR used in the present study.

### 3.2. Biometric Parameters

Of the physiological factors that are possibly linked to the development of ASD, increased HCp and brain growth rates have constantly been implicated as putative early markers of ASD [29]. Although a few studies have focused on physical growth in individuals with ASD [30], little is known about their growth in stature and body proportions. Therefore, all parameters related to growth and HC were compared between the ASD and TDC groups. Furthermore, in order to examine possible differences related to gender, the two groups were divided between male and female, and the data are presented in Table 2. The Wp of ASD boys was significantly higher at six months (*p* = 0.002), nine months (*p* < 0.001), and at 12 months of age (*p* < 0.001) compared to the Wp values of the TDC group, whereas the Wp of ASD females was significantly higher only at nine months (*p* = 0.02) and at twelve months of age (*p* = 0.02). These differences further explain the higher prevalence of ASD in males with excessive weight gain. Table 3 shows that the Hp of the ASD children was significantly higher at three months (*p* = 0.02), as well as at six, nine and twelve months (*p* < 0.001). Comparison of HCp between the ASD and TDC groups showed no differences during the first year of life (Table 4).

### 3.3. Differences in Weight, Height, and Head Circumference Growth Rates

Growth parameters of children in the ASD and TDC groups were compared during their first years of life (in Table 5). Surprisingly, the HC parameters did not differ significantly, although prominent differences in height (Hp) and weight (Wp) were observed (Table 5 and Figure 1). The Wp parameters of the ASD children increased sharply, particularly between one to three months of age, compared to moderate increases of the Wp parameters of the TD children up to six months of age. The putative influence of Wp gain in early infancy was further analyzed by calculating the Wp/Hp ratios. The sharp increase in the Wp/Hp ratio, particularly during the first three months, is attributed to a rapid increase in weight, followed by a prominent increase in height (Table 6 and Figure 2).

Early accelerated brain overgrowth is a prominent contemporary theory of ASD [29]. However, the results described in Table 7 suggest that the changes in HCp are linked to alterations in Wp and Hp between birth and twelve months of age. While a significant acceleration in weight gain continued throughout the first nine months (*p* < 0.001), and the acceleration in height gain continued for six months (*p* = 0.009 at three months; *p* < 0.001 at six months), the acceleration in head circumference could be observed only during the first three months postpartum (*p* = 0.02). It seems, therefore, that measurements of changes only in HCp are insufficient and they should be accompanied by measurements of other skeletal growth parameters.

### 3.4. Age of ASD Diagnosis

The median age of ASD diagnosis in the United States is >4 years [31], although diagnosis can reliably be made in children as young as 18 months of age [32]. van t’Hof and colleagues [33] performed a meta-analysis of studies reported between 1990 and 2012, and they determined that global mean age at diagnosis of autism spectrum disorder ranged from 38 to 120 months with data obtained from over 40 countries. The global mean age at diagnosis was 60.48 months (range: 30.90–234.57 months) based on their analysis. Table 8 demonstrates the mean age of ASD diagnosis among the participants of the present study. Whereas 63.1% of the ASD children were diagnosed before the age of 3, and 14.7% of them were diagnosed as early as in the first year of life, 15.9% of the children were diagnosed for ASD only after age five because they were initially suspected of demonstrating conditions such as ADHD, anxiety, learning disabilities, or other neurodevelopmental disorders. In this respect, the well-organized infrastructure of well-baby clinics in Israel is advantageous in that it enables early identification of ASD as well as identification of early parameters that may serve as predictors of ASD development.

### 3.5. Developmental Milestone Achievement Rates

The initial referral of children with neurodevelopmental delays to either the Child Development Center (CDC), or to a professional therapist (e.g., physiotherapist, occupational therapist, psychiatrist, speech therapist, neurologist, or dietitian), was used to assess primary symptoms of ASD development. Table 9 presents developmental milestone achievement rates of children referred to the CDC. The leading diagnosis of these children was a motor developmental delay (36.7% among the ASD children, and 6.5% in the TDC group (*p* < 0.001)). Notably, a wide range of motor deficits and atypicalities have been reported in ASD children as the earliest presenting symptoms of abnormal brain development [34] (Gowen and Hamilton, 2013). Table 10 summarizes a variety of difficulties in growth and development of children directed to professionals (as indicated by letters of referral). Clearly, more children in the ASD group were referred for a hearing evaluation (*p* < 0.001), to speech therapists (*p* < 0.001), to neurologists (*p* < 0.001), and to psychiatrists (*p* = 0.03). It should be noted that the number of children referred directly to a physiotherapist was similar among the ASD and TDC groups. 

### 3.6. Logistic Regression Analyses

Two logistic regression analyses were performed. The first (Model 1), represented in Table 11 was limited to six months of age, and the second (Model 2), represented in Table 12, included parameters up to 24 months of age. Model 1 ascertained the effects of maternal age, weight and height percentiles at the age of six months, birth weight, gestational age, and gestational diabetes on the likelihood that children would develop ASD. The logistic regression model was statistically significant (*χ*^2^ = 52.014, *p* < 0.001) and explained 4% (Nagelkerke R^2^) of the variance in developing ASD. Children with higher weight and height percentiles at the age of six months were 1.006 and 1.007 times, respectively, more likely to develop ASD. Higher maternal age was also associated with an increased likelihood of developing ASD. Model 2 ascertained the effects of maternal age, weight, and height percentiles at the age of six months, birth weight, gestational age, gestational diabetes, diagnosis of motor delay, and eating problems on the likelihood that children would develop ASD. The logistic regression model was statistically significant (*χ*^2^ = 280.79, *p* < 0.001) and explained 20.2% (Nagelkerke R^2^) of the variance in developing ASD. Children with higher weight and height percentiles at the age of six months were 1.008 and 1.005 times, respectively, more likely to develop ASD. Diagnosis of motor developmental delay, eating problems, and higher maternal age were associated with an increased likelihood of developing ASD.

### 3.7. Physical Comorbidities (Additional Medical Diagnoses)

ASD is frequently associated with other medical conditions that are not specified as part of its diagnosis, such as hearing impairment due to fluid accumulation in the middle ear, sleeping difficulties due to obstruction of the upper airway by adenoid and tonsils hypertrophy, and eating problems [35,36,37]. As shown in Table 13, the prevalence of adenoid enlargement, tonsillectomy, and eating problems in the ASD group was significantly higher in comparison to the TDC group (*p* < 0.001, *p* = 0.008, and *p* < 0.001, respectively). Furthermore, the diagnoses of these comorbidities in the ASD group were made at a younger age (*p* = 0.02). Overall, these results indicate that toddlers with developmental delays (ASD suspects) should be evaluated during their second year of life for adenoid and tonsils hypertrophy, as part of their ASD diagnosis, and be treated as early as possible.

## 4. Discussion

This retrospective cohort study reveals deviations in growth and developmental parameters in infancy that may contribute to the overall variance in developing predictive models for ASD. These parameters include: (A) Extreme upward deviations from trajectories of weight, height, and head circumference percentiles during the first three months of life and continuous excessive increase in weight and height up to twelve months (represented in Table 2, Table 3 and Table 7), which may suggest metabolic and hormonal changes in the body and in the brain leading to brain inflammation and dysfunction in neuronal connectivities [38]. (B) As typical motor development during the first few years of life involves organization of neural networks and connectivities [34], atypical motor development may lead to the formation of alternative neuronal circuits and impede the centers associated with the development of language and speech, coordination, communication, and cognition [39,40].

### 4.1. Metabolic Complications May Lead to ASD Development

The upward deviations from weight trajectories during the first year of life in ASD children may be initiated as a result of genetic and environmental factors [41]. Indeed, a possible link between parental obesity and development of NBDs in their offspring has already been suggested [42,43]. Moreover, deletions in chromosome 16p11.2 were implicated in obesity as well as in NBD [44,45]. Epigenetic obesity was proposed to be linked to DNA modifications (e.g., methylation), affecting the regulation of gene expression [46]. A study of a nationally representative sample of school-aged children linked lesions in the non-dominant right hemisphere and the parietal lobe of children with excessive weight gain to abnormal brain functioning, cognitive impairment in visuo-spatial organization, and general mental ability [47]. Furthermore, Dhanasekara and colleagues [48] have shown in a recent meta-analysis that the risks for cardiometabolic diseases, such as diabetes, hypertension, dyslipidemia, and heart disease were high among children with autism.

The incidence of the “metabolic syndrome” has been shown to parallel the incidence of obesity [49], which, according to the CDC, has an increasing prevalence and is estimated to be 19.7% among children and adolescents aged 2–19 years in 2017–2020 [50]. Human and animal studies have shown that the maternal prenatal “metabolic syndrome” includes increased adiposity and insulin resistance, resulting in an inflammatory state [51,52] with a significant impact on fetal neurodevelopment [53]. Inflammation resulting from a metabolic condition could be a putative mechanism that contributes to the overall variance in ASD [54]. The rapid increase in HCp, following the rapid increase in weight (Table 7), suggests a metabolic process rendering brain inflammation and edema that may be associated with ASD development. This suggestion is corroborated by the studies of Rossignol and Frye [55], as well as Angelidou et al. [56], who have suggested that perinatal stress that induces brain inflammatory responses and increased extra-axial cerebrospinal-fluid (EA-CSF) are associated with a higher risk of developing ASD. The putative connection between excessive weight and brain damage could involve increased body status, such as low-degree inflammation of blood vessels in the brain, since excessive weight was identified as a pro-inflammatory state [57]. In addition, hyperinsulinemia was shown to be associated with impeded glucose metabolism and insulin signaling in several brain regions (e.g., frontal lobes, hippocampus) involved in planning and organizing [58]. Another hormone involved in obesity as well as in ASD development is leptin, found to be elevated in the plasma of obese people [59], and upon placental dysfunction during pregnancy [60]. Indeed, significantly elevated levels of plasma leptin were reported in children with ASD compared to TDC [61].

CSF plays a critical role during brain development and functions throughout life [62] to deliver nutrients and growth factors supporting healthy neural growth and removal of neurotoxins and metabolic waste from neuronal functions [63]. Therefore, in early development and throughout life, CSF production and absorption tend to be balanced. However, perturbations in CSF circulation may impair the clearance of harmful substances that accumulate in the brain and lead to neuro-inflammation [64]. An example of such a condition is idiopathic intracranial hypertension (IIH), a disorder typified by elevated intracranial pressure with an estimated incidence of 15–19 cases per 100,000 among overweight or obese 20–44-year-old women [65,66]. Overall, excess weight is an important risk factor signifying the development of IIH among post-pubertal children or adults [67]. Moreover, the high percentage of IIH among girls that gain weight rapidly at menarche may explain, in part, its sudden increase. Recently, increased EA-CSF, originating in infancy and observed through preschool age, found in three independent cohort studies of children with ASD, was proposed as a putative ASD marker [68,69,70].

Courchesne et al. [71] reported in a longitudinal study that 59% of the infants they studied had demonstrated accelerated HC growth trajectories (>2.0 SDs) from birth up to 14 months of age, whereas only 6% of the TDC infants followed these accelerated growth trajectories. In a recent prospective study using brain MRI scans of high-risk infants, Shen and co-workers concluded that increased EA-CSF at six months of age was a significant predictor for later developing ASD [69]. Thus, early accelerated brain enlargement could be a contributing factor to ASD development. It is worth noting that upward deviations from weight and height growth trajectories of infants, who were later diagnosed for ASD, have been previously reported in a retrospective study [25]. In an attempt to explain the link between growth dysregulation and ASD development, Green and colleagues [30] have proposed two mechanisms: A) a connective tissue disorder, frequently associated with increased height and disproportionate body ratios and B) dysregulation of the hypothalamic-pituitary-adrenal (HPA) axis that regulates growth hormones. Notably, the clinical significance of larger brain volumes in ASD toddlers is still controversial. The present study reveals that, except for the initial rapid increase, no significant differences in HCp between the ASD and TDC groups could be observed after the first three months of life (Table 7), suggesting that brain volume by itself is not a suitable predictor of ASD development, as also corroborated by a recent study [72]. The present study supports the theory that ASD is associated more with dysregulated growth in general, rather than to merely a dysregulation of neuronal growth in the brain. Still, it remains unclear whether this non-regulated growth could serve as a biomarker of ASD development.

### 4.2. Atypical Motor Development May Be Associated with ASD

Motor difficulties are the principal reasons for toddlers’ visits to Child Developmental Centers, and so the first professional to meet the child is likely a physiotherapist. Surprisingly, however, no difference between the ASD and TDC groups was observed, implying that motor delay and referral to a physiotherapist cannot serve as a predictive factor specific for ASD. Instead, a dramatic increase in referrals for assessment and treatment of deformational plagiocephaly (DP), reported over the last two decades, might be related to the “Back to Sleep” guidelines as a preventive measure of Sudden Infant `death Syndrome (SIDS) [73]. Although considered a medically benign condition, DP may be associated with developmental difficulties, suggesting that children with this condition should be routinely followed until school age for developmental delays or deficits [74]. Numerous authors have theorized about the relation between sleeping position and ASD [75,76,77]. Another prospective study on infants at risk has shown significantly poorer fine and gross motor scores at 36 months of age in children diagnosed at a later age for ASD [78]. These results have suggested that early motor difficulties may predict a forthcoming NBD. A wide range of motor delays and deficits that constitute the core of s have also been reported in ASD and in ADHD individuals, and there is a link between early deviations from motor developmental trajectories and psychological outcomes in childhood [25,79]. Overall, although motor delay problems cannot serve as a single risk factor predicting forthcoming ASD, all babies may benefit from motor intervention for the prevention of asymmetry, plagiocephaly, torticollis and motor delay from the first month after birth.

### 4.3. Study Limitations

The study described here is a post hoc cohort study of data-mining that makes use of the outcomes of over 20 years of data-gathering from a database connected to one of the major HMOs in the State of Israel. The instant report does not cover all potentially influencing factors because they could not possibly have been identified at the outset. There were questions in the database about topics like vaginal vs. cesarian section delivery or breastfeeding vs. formula feeding. Although they were not accessible, potential confounding factors, such as more specific pregnancy information and racial or ethnic practice variations, will undoubtedly be the foundation for future research. Due to the fact that the MHO primarily served people from middle class backgrounds, concerns relating to poverty that were absent from the original data source are now incorporated. Another disadvantage is that variations in head circumference, weight, and height may be due to disparities between autistic and non-autistic people rather than problems that need to be rectified to “prevent” people from “becoming” autistic. Given the multifactorial nature of our current understanding of the etiology of ASD and the search for early markers, the small proportion of the variance still being significant can potentially contribute to the development of biomarkers to bring the age of identification lower and for interventions to commence earlier.

One possible approach to increase the variance is to divide the infants into three groups according to their birth weight (large (LGA), appropriate (AGA), and small (SGA) gestational age) and follow the changes in weight, height, and head circumference growth rates along the first year of life. The possibility that the LGA infant went through a rapid weight gain and increased HC during the last trimester should be raised. We could not tell from our study if the metabolic and motoric disturbances are primary or secondary to ASD, but they could be considered as two early predictive environmental factors. It is our intention to work with MHO big data sets in the near future to seek further early signs of predictive factors of ASD.

## 5. Conclusions

Rapid weight and height gains at early infancy followed by an excessive increase in head circumference are significantly related to ASD development, and thus might serve as predicting factors. The correlation between rapid increase in weight and the development of ASD suggests that prevention of intrauterine and infant rapid weight gain might be significant in diminishing the number of children with ASD. Since brain plasticity in infancy enables manipulations to strengthen motor development and the corresponding neural functions, the sooner a risk factor is recognized in an infant, the more effective intervention is likely to be. Early intervention programs to prevent rapid weight gain and guidelines for motor development during infancy, when the plasticity of the brain is high, may assist in reducing the number of children with ASD.

## Figures and Tables

**Figure 1 brainsci-13-01374-f001:**
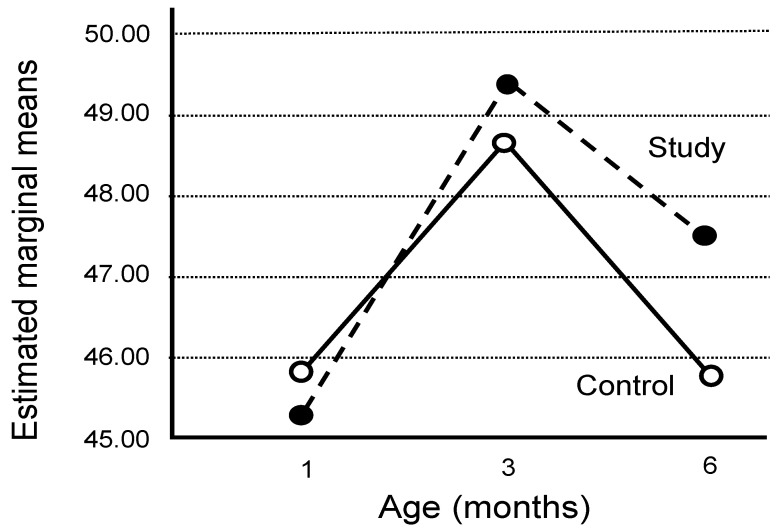
Difference in rates of weight percentile changes during the first 6 months between the ASD and TDC groups.

**Figure 2 brainsci-13-01374-f002:**
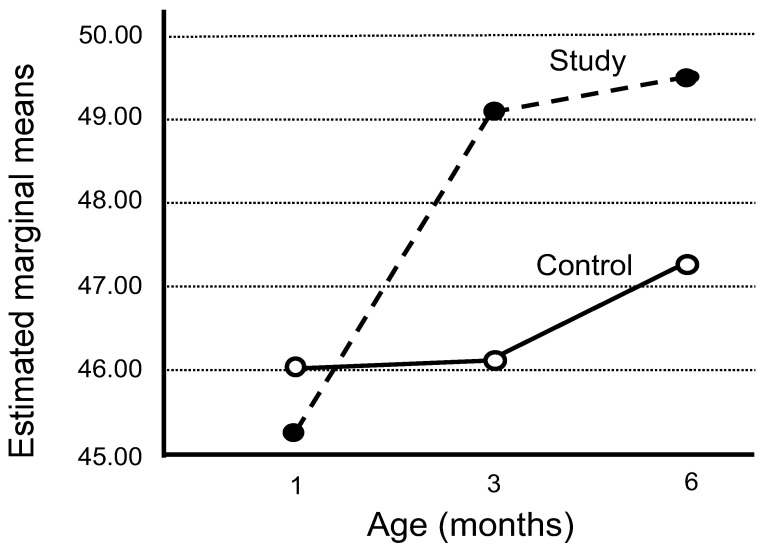
Difference in rates of changes in weight to height percentile ratio (Wp/Hp) during the first 6 months between the ASD and TDC groups.

**Table 1 brainsci-13-01374-t001:** Demographic characteristics of the study groups.

		TDC	ASD	
		N (%)	N (%)	*p*
Gender	male	916 (82.9)	916 (82.9)	1
	female	189 (17.1)	189 (17.1)	
Socioeconomic (%)	low	138 (12.5)	161 (14.6)	0.33
	medium	477 (43.2)	473 (42.8)	
	high	490 (44.3)	471 (42.6)	
District (%)	North	67 (6.1)	67 (6.1)	1
	Sharon	335 (30.3)	335 (30.3)	
	South	315 (28.5)	315 (28.5)	
	Center	143 (12.9)	143 (12.9)	
	Shfela and Jerusalem	245 (22.2)	245 (22.2)	
Gestational age (weeks) (mean ± SD)		39.12 ± 1.28	39.22 ± 1.22	0.07
Birth weight (Kg)		3.32 ± 0.43	3.2764 ± 0.46	0.007
Maternal age (years) (mean ± SD)		32.11 ± 5.08	32.83 ± 5.24	0.001
Maternal gestational diabetes (%)	yes	49 (4.4)	68 (6.2)	0.08
Alcohol abuse during pregnancy (%)	yes	41 (3.7)	34 (3.1)	0.48
Smoking during pregnancy (%)	yes	60 (9)	66 (8.6)	0.89
Fertility treatments (%)	infertility diagnosis	144 (55.8)	168 (57.1)	0.86
	non-IVF treatments	108 (41.9)	121 (41.2)	
	IVF treatments	6 (2.3)	5 (1.7)	

**Table 2 brainsci-13-01374-t002:** Differences in weight percentiles between the study groups during the first year of life.

Weight Percentiles		GROUP				
		TDC		ASD		
Age (Months)	Gender	N	(Mean ± SD)	N	(Mean ± SD)	*p*-Value
1	male	841	46.68 ± 0.90	853	45.68 ± 1.96	0.43
	female	178	43.22 ± 0.90	169	44.30 ± 2.01	0.7
3	male	910	54.07 ± 0.88	898	56.10 ± 0.88	0.1
	female	186	52.97 ± 1.95	181	55.103 ± 1.97	0.44
6	male	803	45.90 ± 0.97	795	50.27 ± 0.98	0.002
	female	156	47.28 ± 2.21	162	51.33 ± 2.17	0.19
9	male	741	38.88 ± 1.07	793	44.24 ± 0.98	<0.001
	female	156	40.31 ± 2.21	159	47.16 ± 2.19	0.02
12	male	817	36.47 ± 0.97	824	44.05 ± 0.97	<0.001
	female	162	37.32 ± 2.19	162	44.30 ± 2.19	0.02

**Table 3 brainsci-13-01374-t003:** Differences in height percentiles between the study groups during the first year of life.

Height Percentiles	GROUP				
	TDC		ASD		
Age (Months)	N	(Mean ± SD)	N	(Mean ± SD)	*p*-Value
1	836	46.09 ± 24.36	825	47.51 ± 25.18	0.24
3	987	56.77 ± 25.44	958	59.38 ± 24.67	0.02
6	862	58.41 ± 24.71	844	62.67 ± 24.58	<0.001
9	762	53.31 ± 25.05	802	58.69 ± 24.84	<0.001
12	819	50.04 ± 25.86	819	56.72 ± 25.79	<0.001

**Table 4 brainsci-13-01374-t004:** Differences in head circumference percentiles between the study groups during the first year of life.

Head Circumference					
	TDC		ASD		
Age (Months)	N	(Mean ± SD)	N	(Mean ± SD)	*p*-Value
1	1001	37.58 ± 21.01	1003	37.08 ± 22.83	0.61
3	1090	39.31 ± 23.84	1074	39.31 ± 25.11	0.699
6	933	42.69 ± 25.18	935	43.40 ± 26.68	0.55
9	854	44.61 ± 26.75	928	46.21 ± 27.37	0.21
12	959	47.20 ± 26.79	969	48.72 ± 27.90	0.22

**Table 5 brainsci-13-01374-t005:** Changes in weight percentiles during the first 6 months postpartum.

Weight Percentiles	GROUP				
	TDC		ASD		
Age (Months)	N	(Mean ± SD)	N	(Mean ± SD)	*p*-Value
1	893	46.5 ± 24.86	888	44.99 ± 26.97	0.21
3	893	54.17 ± 25.88	888	56.12 ± 26.9	0.12
6	893	46.24 ± 27.14	888	50.91 ± 28.42	<0.0001

**Table 6 brainsci-13-01374-t006:** Weight to height percentile ratios (Wp/Hp) of the study groups during the first 6 months postpartum.

Wp/Hp	GROUPS				
	CONTROL		ASD		
1	684	45.99 ± 26.6	691	45.17 ± 27.05	0.57
3	684	46.06 ± 28.12	691	49.07 ± 28.75	0.05
6	684	47.23 ± 28.46	691	48.37 ± 28.73	0.144

**Table 7 brainsci-13-01374-t007:** Differences between the study groups in head circumference, weight and height percentiles.

Months (m)	TDC	ASD	*p*-Value
**Head Circumference Percentile**			
Gap between			
3 to 1 m	1.76 ± 14.2	3.18 ± 14.36	0.02
Gap between			
6 to 3 m	3.84 ± 11.4	4.22 ± 11.71	0.48
Gap between			
9 to 6 m	2.78 ± 8.96	3.13 ± 10.25	0.47
Gap between			
12 to 9 m	2.83 ± 11.33	2.6 ± 12.28	0.69
**Weight percentile**			
Gap between			
3 to 1 m	7.79 ± 20.12	11.16 ± 21.32	<0.001
Gap between			
6 to 3 m	−7.99 ± 14.37	−5.2 ± 14.16	<0.001
Gap between			
9 to 6 m	−6.85 ± 11.24	−5.19 ± 12.1	0.004
Gap between			
12 to 9 m	−1.96 ± 11.98	−1.55 ± 12.63	0.49
**Height percentile**			
Gap between			
3 to 1 m	12.16 ± 18.04	14.55 ± 17.85	0.009
Gap between			
6 to 3 m	1.24 ± 16.01	4.56 ± 16.2	<0.001
Gap between			
9 to 6 m	−5 ± 13.81	−3.82 ± 14.28	0.136
Gap between			
12 to 9 m	−3.17 ± 13.7	−3.12 ± 14.96	0.95

**Table 8 brainsci-13-01374-t008:** Distribution of ASD children according to age of diagnosis.

Age of ASD Diagnosis	Number of Children	%
0–1 years	162	14.7
1–2 years	298	27
2–3 years	236	21.4
3–4 years	139	12.6
4–5 years	94	8.5
>5 years	176	15.9
Total	1105	100

**Table 9 brainsci-13-01374-t009:** Developmental milestone achievements in motor, communication, speech, and language based on referral letters to specialists.

	TDC (N = 417) (Mean ± SD)	ASD (N = 1091) (Mean ± SD)	OR (95% CI)	*p*-Value
**Age (years) Delay**	3.62 ± 3.6	2.39 ± 1.48		<0.001
**Speech and Language, n (%)**	37 (3.3)	316 (28.6)	11.56 (8.12–16.45)	<0.001
**Age of first diagnosis (years)**	2.48 ± 0.44	2.32 ± 0.48		0.04
**Motor, n (%)**	72 (6.5)	406 (36.7)	8.33 (6.3–10.89)	<0.001
**Age of first diagnosis (years)**	1.3 ± 0.7	1.8 ± 0.73		<0.001
**Communication, n (%)**	3 (0.3)	144 (13)	55 (17.48–173.24)	<0.001
**Age of first diagnosis (years)**	2.24 ± 0.42	2.26 ± 0.49		0.94

**Table 10 brainsci-13-01374-t010:** Differences between the ASD and TDC groups in referrals to specialists.

Referral	TDC (N = 422) n (%)	ASD (N = 912) n (%)	OR (95% CI)	*p*-Value
**Hearing Center (%)**	255 (60.4)	725 (79.5)	2.53 (1.97–3.26)	<0.001
**age of first referral (months)**	21.27 ± 9.82	20.85 ± 7.66		0.12
**Dietitian**	71 (16.8)	125 (13.7)	0.78 (0.57–1.07)	0.13
**age of first referral (months)**	17.7 ± 9.52	17.39 ± 9		0.95
**Child Development Center**	106 (25.1)	484 (53.1)	3.71 (2.61–4.35)	<0.001
**age of first referral (months)**	16.24 ± 10.25	20.64 ± 9.18		<0.001
**Occupational therapist**	7 (1.7)	31 (3.4)	2.08 (0.91–4.77)	0.07
**age of first referral (months)**	25.66 ± 8.72	28.25 ± 6.87		0.73
**Psychiatrist**	2 (0.5)	18 (2)	4.22 (0.97–18.3)	0.03
**age of first referral (months)**	19 ± 9.89	29.56 ± 7		0.11
**Physiotherapist**	79 (18.7)	157 (17.2)	0.9 (0.67–1.21)	0.5
**age of first referral (months)**	6.83 ± 7.64	8.76 ± 9.21		0.29
**Speech therapist**	62 (14.7)	208 (22.8)	1.71 (1.25–2.34)	<0.001
**age of first referral (months)**	28.34 ± 7.39	27.38 ± 5.83		0.03
**Neurologist**	46 (10.9)	203 (22.3)	2.34 (1.66–3.3)	<0.001
**age of first referral (months)**	15.1 ± 11.06	20.59 ± 10.11		0.002

**Table 11 brainsci-13-01374-t011:** Model 1. Logistic Regression Analysis.

Variables in the Equation									
		B	S.E.	Wald	df	Sig.	Exp (B)	95% C.I. for EXP (B)	
								Lower	Upper
	Weight percentile, age 6 months	0.006	0.002	7.649	1	0.006	1.006	1.002	1.011
	Birth weight, Kg	−0.585	0.136	18.511	1	<0.001	0.557	0.427	0.727
	Gestational age, weeks	−0.034	0.044	0.590	1	0.442	0.967	0.887	1.054
	Mother age, years	0.026	0.010	7.022	1	0.008	1.026	1.007	1.046
	Height percentile, age 6 months	0.007	0.002	7.465	1	0.006	1.007	1.002	1.012
	Gestational diabetes	0.241	0.217	1.228	1	0.268	1.272	0.831	1.949
	Constant	1.654	1.648	1.008	1	0.315	5.229		

**Table 12 brainsci-13-01374-t012:** Model 2. Logistic Regression Analysis.

Variables in the Equation									
		B	S.E.	Wald	df	Sig.	Exp (B)	95% C.I. for EXP (B)	
								Lower	Upper
	Weight percentile, age 6 months	0.008	0.002	9.251	1	0.002	1.008	1.003	1.012
	Birth weight, Kg	−0.492	0.145	11.457	1	<0.001	0.611	0.460	0.813
	Gestational age, weeks	−0.001	0.047	0.000	1	0.986	0.999	0.911	1.095
	Mother age, years	0.024	0.010	5.100	1	0.024	1.024	1.003	1.045
	Height percentile, age 6 months	0.005	0.003	3.793	1	0.051	1.005	1.000	1.010
	Gestational diabetes	0.279	0.231	1.456	1	0.228	1.322	0.840	2.079
	Motoric delay	2.035	0.159	163.777	1	<0.001	7.654	5.604	10.453
	Eating disorders	1.202	0.391	9.442	1	0.002	3.328	1.546	7.166
	Constant	−0.213	1.770	0.015	1	0.904	0.808		

**Table 13 brainsci-13-01374-t013:** Various medical diagnoses (comorbidities) and age of diagnosis.

Medical Diagnosis	TDC N = 1105	ASD N = 1105	OR (95% CI)	*p*-Value
**Neonatal Jaundice (%)**	184 (23.4)	174 (19.7)	0.8 (0.63–1.01)	0.07
**age of diagnosis (months)**	0.03 ± 0.2	0.43 ± 3.97		0.22
**Allergies (%)**	180 (18.6)	180 (16.8)	0.88 (0.7–1.1)	0.28
**age of diagnosis (months)**	19.51 ± 16.51	18.23 ± 16.27		0.44
**Adenoids Hypertrophy (%)**	130 (13.4)	214 (19.9)	1.6 (1.2–2.03)	<0.001
**age of diagnosis (months)**	31.74 ± 14.13	28.72 ± 11.46		0.02
**Asthma (%)**	226 (23.3)	168 (15.7)	0.6 (0.48–0.76)	<0.001
**age of diagnosis (months)**	19.53 ± 13.57	22.67 ± 14.39		0.01
**Anemia (%)**	104 (10.7)	97 (9)	0.8 (0.61–1.1)	0.2
**age of diagnosis (months)**	19.66 ± 12.05	19.61 ± 13.94		0.55
**Tonsillectomy (%)**	51 (5.3)	168 (15.7)	1.6 (1.12–2.29)	0.008
**age of diagnosis (months)**	36.56 ± 13.17	33.34 ± 10.13		0.1
**Eating problems (%)**	11(1.1)	43 (4)	3.6 (1.8–7)	<0.001
**age of diagnosis (months)**	22.63 ± 14.36	30.2 ± 14.3		0.11
**Failure to Thrive (%)**	72(7.4)	71 (6.6)	0.88 (0.62–1.24)	0.47
**age of diagnosis (months)**	13.3 ± 8.59	12.9 ± 9.42		0.42

## Data Availability

The datasets for this study are part of the Big Data Repository of the Maccabi HMO; questions about the database and the data from the current study may be addressed to Liat Lev-Shalem, member of the MAROM program.

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
