# Peer review of "Factors in Infancy That May Predict Autism Spectrum Disorder"

_brainsci, 2023, doi:10.3390/brainsci13101374_

Round 1
Reviewer 1 Report
I thank the editor for the possibility to review the present paper.
Following my suggestions to improve the paper.
(1) authors refer only to one study on screening tools. I believe that systematic reviews should be more appropriate to cite.
(2) I believe that the aims of the study should be reported clearly.
(3) Regarding the discussion section, the results are well described. However, there are several typos that require correction. In study limitation section, future directions should be extended. Finally, the conclusion section should include a detailed discussion of the possible implications of the results.
Several typos should be corrected. Furthermore, to improve the readability of the paper, minor revision of English language is required.
Author Response
Following my suggestions to improve the paper.
1. The authors refer only to one study on screening tools. I believe that systematic reviews should be more appropriate to cite.
1. We thank the reviewer for the helpful points. Systematic reviews have now been included expanding the literature on screening tools for the early detection of ASD with their current attendant problems as well as possibilities doe effective use. These additions are reflected between lines 63-65 in the marked copy.
2. I believe that the aims of the study should be reported clearly.
2. Study aims have been clearly stated and included both in the body of the ms. (lines 123-126) as well as in the abstract (lines11-13).
3. Regarding the discussion section, the results are well described. However, there are several typos that require correction.
3. We apologize for the typos and the ms. has been reread carefully to correct those types of problems throughout.
4. In study limitation section, future directions should be extended. Finally, the conclusion section should include a detailed discussion of the possible implications of the results.
4. Future directions in the Limitation Section have been extended (lines 585-587) and the conclusions have been expanded with the implications of the results.
5. Comments on the Quality of English Language. Several typos should be corrected. Furthermore, to improve the readability of the paper, minor revision of English language is required.
5. The paper has been read and reread and we think that the typos, for which we apologize, have been addressed and corrected. We have also reviewed the text to increase the ease of “readability” as the reviewer had suggested.
Reviewer 2 Report
The authors of this work tried to identify common early factors in children later diagnosed with ASD.
The results of the study presented in the article are important for early prediction of ASD manifestation. The correctness of the data collection procedure for the study is beyond doubt, as is the correctness of the statistical analysis of the data.
However two logistic regression models, including early symptoms of ASD, explain a very small proportion of the variance in the formation of ASD (4% of the first and 20% of the second). Nevertheless, the early indicators of ASD identified in this study may be useful for the practice of building better models for predicting ASD.
In the abstract: «This study sought to identify common early factors in children later diagnosed with ASD»(s. 11-12). «Early intervention programs to prevent rapid weight gain and guidelines for motor development during infancy, when the brain is highly plastic, can assist in reducing the number of children with ASD» (s.22-23) - such cause-and-effect interpretations a la speculations of the discovered statistical relationships should be put into the Discussion section.
It is very doubtful that the factors that influence the occurrence of ASD have been found. The results of this study are the identified early indicators of the subsequent manifestation of ASD. The problem is: "after" something does not always mean "because" of it. It is necessary to correct the description of the results obtained. The authors should be more careful in interpreting statistical relationships.
In my opinion, the English language is quite correct, maybe minimal corrections are required.
Author Response
1. The authors of this work tried to identify common early factors in children later diagnosed with ASD.
The results of the study presented in the article are important for early prediction of ASD manifestation. The correctness of the data collection procedure for the study is beyond doubt, as is the correctness of the statistical analysis of the data.
1. We thank the reviewer for affirming the appropriateness of the statistical methodology
2. However two logistic regression models, including early symptoms of ASD, explain a very small proportion of the variance in the formation of ASD (4% of the first and 20% of the second). Nevertheless, the early indicators of ASD identified in this study may be useful for the practice of building better models for predicting ASD.
2. The reviewer is correct when pointing out the proportion of the variance explained in attempting to build a predictive model. Given the multifactorial nature of our current understanding of the etiology of ASD and the search for early markers, the small proportion of the variance still being significant can contribute to the development of biomarkers to bring the age of identification lower and for interventions to commence earlier. It is, therefore, both a limitation and a contribution at the same time. The reviewer’s point, therefore, is well taken. We have added the reviewer’s point and our response to the discussion section of the ms.
3. In the abstract: «This study sought to identify common early factors in children later diagnosed with ASD»(s. 11-12).
3. Qualifications have been added to the abstract (lines 12-13 of the marked copy)
4. «Early intervention programs to prevent rapid weight gain and guidelines for motor development during infancy, when the brain is highly plastic, can assist in reducing the number of children with ASD» (s.22-23) - such cause-and-effect interpretations a la speculations of the discovered statistical relationships should be put into the Discussion section.
4. We have removed lines 25-27 and placed them in the discussion/conclusion section now between lines 585-587/597-599.
5. It is very doubtful that the factors that influence the occurrence of ASD have been found. The results of this study are the identified early indicators of the subsequent manifestation of ASD. The problem is: "after" something does not always mean "because" of it. It is necessary to correct the description of the results obtained. The authors should be more careful in interpreting statistical relationships.
5. We completely agree with the reviewer when indicating that correlation does not imply causality and have reviewed the manuscript again to correct any such misleading statements. (e.g., lines 448-450, 409-411, 413, 437, 499-505, 527, 412-414, 433-434 in the marked text).
Round 2
Reviewer 2 Report
I do not see any obstacles to recommend this manuscript for publication.